# Acne Transcriptomics: Fundamentals of Acne Pathogenesis and Isotretinoin Treatment

**DOI:** 10.3390/cells12222600

**Published:** 2023-11-10

**Authors:** Bodo C. Melnik

**Affiliations:** Department of Dermatology, Environmental Medicine and Health Theory, University of Osnabrück, 49069 Osnabrück, Germany; melnik@t-online.de; Tel.: +49-5241988060

**Keywords:** acne, apoptosis, FoxO1, FoxO3, isotretinoin, mTORC1, p53, pathogenesis, therapy, transcriptomics

## Abstract

This review on acne transcriptomics allows for deeper insights into the pathogenesis of acne and isotretinoin’s mode of action. Puberty-induced insulin-like growth factor 1 (IGF-1), insulin and androgen signaling activate the kinase AKT and mechanistic target of rapamycin complex 1 (mTORC1). A Western diet (hyperglycemic carbohydrates and milk/dairy products) also co-stimulates AKT/mTORC1 signaling. The AKT-mediated phosphorylation of nuclear FoxO1 and FoxO3 results in their extrusion into the cytoplasm, a critical switch which enhances the transactivation of lipogenic and proinflammatory transcription factors, including androgen receptor (AR), sterol regulatory element-binding transcription factor 1 (SREBF1), peroxisome proliferator-activated receptor γ (PPARγ) and signal transducer and activator of transcription 3 (STAT3), but reduces the FoxO1-dependent expression of GATA binding protein 6 (GATA6), the key transcription factor for infundibular keratinocyte homeostasis. The AKT-mediated phosphorylation of the p53-binding protein MDM2 promotes the degradation of p53. In contrast, isotretinoin enhances the expression of p53, FoxO1 and FoxO3 in the sebaceous glands of acne patients. The overexpression of these proapoptotic transcription factors explains isotretinoin’s desirable sebum-suppressive effect via the induction of sebocyte apoptosis and the depletion of BLIMP1(+) sebocyte progenitor cells; it also explains its adverse effects, including teratogenicity (neural crest cell apoptosis), a reduced ovarian reserve (granulosa cell apoptosis), the risk of depression (the apoptosis of hypothalamic neurons), VLDL hyperlipidemia, intracranial hypertension and dry skin.

## 1. Introduction

In recent years, tremendous scientific progress has been made in unraveling transcriptional regulation in acne vulgaris. This review provides updated information on acne transcriptomics, allowing for deeper insights into the pathogenesis and treatment of acne. Acne vulgaris is a very common chronic, inflammatory skin disorder with a complex pathogenesis. Four factors play vital roles in the pathophysiology of acne: hyperseborrhea and dysseborrhea, the altered keratinization of the pilosebaceous duct (comedogenesis), effects mediated by *Cutibacterium acnes* (*C. acnes*) and Th17-cell-driven inflammation [1,2]. 

Isotretinoin is the most efficient anti-acne drug, improving all major factors of the pathogenesis of acne. This review focuses on three key transcription factors, p53, FoxO1 and FoxO3, whose suppression plays an important role in the pathogenesis of acne, whereas their upregulation via isotretinoin explains isotretinoin’s beneficial and adverse effects in the treatment of acne.

## 2. Growth Factor Signaling in Acne: Activating the Kinase AKT 

Various growth factors induce the pathogenesis of acne. Their input signals converge in the activation of the kinase AKT (protein kinase B), which modifies the activity and expression of important transcription factors promoting the disease. 

### 2.1. Insulin-like Growth Factor 1

Insulin-like growth factor 1 (IGF-1) is a potent mitogen [3]. IGFs bind specifically to the IGF-1 receptor (IGF1R) on the cell surface of a targeted tissue [4]. During puberty, the increasing production of growth hormone (GH) leads to the activation of the GH/IGF-1 axis [5]. IGF-1 is the key hormone in puberty [6] and promotes sexual differentiation and linear bone growth [7]. Acne becomes manifest at a relatively late stage of puberty, at or shortly after peak height velocity in boys and menarche in girls [8]. Deplewski and Rosenfield [9] emphasized the important role of IGF-1 in the development of the pilosebaceous unit. Increased serum IGF-1 levels correlate with acne lesion counts in females [10] and increased sebum secretion rates in male acne patients [11]. IGF-1 is overexpressed in the epidermis and pilosebaceous units of acne patients compared to controls [12]. A Western diet, especially the consumption of hyperglycemic carbohydrates [13], milk and yogurt [14,15], has been associated with IGF-1-mediated acne pathogenesis [16,17]. Individuals with Laron syndrome, who exhibit a congenital IGF-1 deficiency, are of short stature and never develop acne [18] unless therapeutically substituted with recombinant IGF-1 [19]. 

### 2.2. IGF-1-PI3K-AKT-Mediated Downregulation of FoxO1 and FoxO3

After the binding of IGF-1 to its receptor, phosphoinositide 3-kinase (PI3K) is activated, resulting in the activation of the kinase AKT (also known as protein kinase B), a key checkpoint of growth network regulation [20]. Among many regulatory effectors, AKT phosphorylates and thereby inactivates the transcription factors FoxO1 and FoxO3, which are predicted to play a key role in the pathogenesis of acne [21]. After AKT-mediated phosphorylation, nuclear FoxO1 is extruded into the cytoplasm, which was confirmed in IGF-1-stimulated SZ95 sebocytes in vitro [22]. FoxO1 acts as a negative nuclear coregulator of lipogenic transcription factors including androgen receptor (AR) [23], sterol regulatory element-binding transcription factor 1 (SREBF1) [24] and peroxisome proliferator-activated receptor-γ (PPARγ) [25] and of the pro-inflammatory signal transducer and activator of transcription 3 (STAT3) [26]. 

FoxO1 binds to STAT3 and prevents STAT3 from interacting with the SP1.POMC promoter complex, consequently inhibiting the STAT3-mediated action of leptin. As shown in SZ95 sebocytes, leptin promotes a proinflammatory lipid profile and induces inflammatory pathways in human SZ95 sebocytes [27].

Thus, increased IGF-1 signaling via the AKT-mediated suppression of FoxO1 enhances the transcriptional activity of AR, SREBF1, PPARγ and STAT3, crucial transcription factors which promote sebaceous lipogenesis [28,29,30]. In fact, IGF-1 has been shown to induce SREBF1 expression and lipogenesis in SEB-1 sebocytes via the activation of the PI3K/AKT pathway [30], whereas a low glycemic diet, which decreases IGF-1 serum levels [12], reduces SREBF1 expression in the sebaceous glands (SGs) of acne patients [31] (Figure 1).

### 2.3. IGF-1-IGF1R-PI3K-AKT-MDM2-Mediated Downregulation of P53

IGF-1/IGF1R/PI3K/AKT activation inhibits the activity of p53 [32]. AKT activated by exogenous IGF-1 promotes the phosphorylation of the p53-binding protein mouse double minute 2 (MDM2). This phosphorylation increases the ability of MDM2 to degrade p53 [32]. MDM2 promotes cell survival and cell cycle progression by inhibiting p53. To regulate p53, MDM2 must gain nuclear entry, thereby diminishing cellular levels of p53, and decreases p53’s transcriptional activity [32]. Phosphorylated MDM2 finally promotes the proteasomal degradation of p53 [33,34]. Thus, IGF-1-AKT signaling lowers the level of nuclear p53, which was recently suggested to play a key role in the pathogenesis and treatment of acne [2,35] (Figure 1).

### 2.4. IGF-1-IGF1R-PI3K-AKT-Mediated Activation of mTORC1

Mechanistic target of rapamycin complex 1 (mTORC1) is a key growth factor and nutrient-dependent regulatory kinase orchestrating cell proliferation and anabolism and inhibiting autophagy [36,37]. Activated AKT phosphorylates tuberin (TSC2). TSC2 is inactivated via AKT-dependent phosphorylation, which destabilizes TSC2 and disrupts its interaction with hamartin (TSC1) [38]. The TSC1-TSC2 (hamartin-tuberin) complex, through its GAP (GTPase-activating protein) activity toward the small G-protein RHEB (Ras homolog protein enriched in brain) is a critical negative regulator of mTORC1 [39]. Thus, the AKT-mediated phosphorylation of TSC2 plays a key role in the growth factor-stimulated activation of mTORC1, which is predicted to be of fundamental importance in the pathogenesis of acne [40,41]. Indeed, increased levels of expression of mTOR and mTORC1 activation have been detected in the epidermis and SGs of acne patients [42,43,44]. The consumption of a Western diet over-stimulates insulin/IGF-1/AKT/mTORC1 signaling [45]. Recent evidence indicates that milk is a highly specialized mammalian endocrine-signaling system which promotes mTORC1-dependent translation for postnatal growth [46]. mTORC1 has a special impact in regulating the protein translation of specific transcription factors involved in the pathogenesis of acne, including SREBFs, PPARγ, STAT3 and hypoxia-inducible factor-1α (HIF-1α) [47]. Further evidence obtained in adipocytes underlines the role of mTORC1 in leptin biosynthesis at the level of translation [48] (Figure 1).

### 2.5. Insulin-INSR-PI3K-AKT-Mediated Activation of mTORC1

Acne has been associated with syndromes associated with insulin resistance, especially polycystic ovary syndrome (PCOS) [49], which results in hyperinsulinemia [50]. Insulin and IGF-1 are sister growth hormones, and insulin has been shown to bind and activate both insulin receptor (INSR) and IGF1R. Both INSR and IGF1R also elicit common downstream signaling with the phosphorylation of a family of INSR substrates and the activation of the PI3K-AKT-mTORC1 pathway [51]. Milk exhibits a high insulinemic index [52]. High intakes of milk, but not meat, increase serum insulin and insulin resistance, as shown in 8-year-old boys [53]. Thus, milk consumption stimulates insulin and IGF-1 signaling [45], a meaningful mechanism to stimulate mTORC1-driven postnatal growth and anabolism [46]. In contrast, metformin therapy lowers insulin resistance [54] and thus reduces insulin signaling toward SGs. Mirdamadi et al. [22] convincingly demonstrated that both insulin and IGF-1 enhance the PI3K/AKT pathway in SZ95 sebocytes in vitro.

### 2.6. FGFR2-PI3K-AKT-Mediated Activation of mTORC1

Patients with congenital fibroblast growth factor receptor 2 (FGFR2) gain-of-function mutations (S252W or P253R FGFR2)—either germline (Apert syndrome) [55,56,57,58] or segmental somatic (acne nevus of Munro) [59,60,61,62,63]—develop acne during puberty and respond to isotretinoin treatment [64]. FGFR2 is a tyrosine kinase receptor which, like INSR or IGF1R, activates multiple pathways, including PI3K/AKT signaling [55]. Gain-of-function FGFR2 signaling thus adds on IGF1R and INSR signaling in puberty, enhancing the AKT-mTORC1 pathway and causing the acne in Apert syndrome and acne nevus of Munro (Figure 1).

### 2.7. Androgen Receptor Signaling Converges with the PI3K-Mediated Activation of AKT

Decades ago, acne was believed to be primarily an androgen-driven disease. Although the impact of androgens on the pathogenesis of acne is well accepted [2,28,65], recent observations underline a complex, synergistic crosstalk between IGF-1, insulin and androgens in the activation of AKT and mTORC1. Androgens operate via ligand binding to ARs. IGF-1 plays a critical role in the synthesis of adrenal and gonadal androgens [2,9,66,67,68,69,70,71] and promotes the 5α-reductase-mediated conversion of testosterone into dihydro-testosterone (DHT) [72], the high-affinity ligand of ARs [73,74]. Androgens induce sebaceous differentiation in sebocytes expressing a stable, functional AR [75]. Notably, two androgen response elements (AREs) have been identified in the upstream promoter of the *IGF1* gene [76]. It has been shown in prostate cancer cells that androgens also upregulate IGF1R expression [77]. DHT increases the association of mTOR with RICTOR instead of RAPTOR, consistent with the selective activation of mechanistic target of rapamycin complex 2 (mTORC2) by DHT [78]. Like PI3K, the growth-promoting kinase mTORC2 phosphorylates and activates AKT and thereby contributes to the nuclear extrusion of FoxO1 [78]. Thus, the insulin/IGF-1-PI3K signaling pathway and the androgen-mTORC2 signaling pathway converge in the activation of AKT (Figure 1). The AKT-mediated nuclear extrusion of FoxO1, which is a negative coregulator of AR, further enhances the transcriptional activity of AR [23]. In addition, AKT phosphorylates ARs, enhancing AR transactivation and the nuclear stability of AR [79]. The AKT-MDM2-mediated suppression of p53 [32,33,34] will further enhance AR signaling because the expression of ARs is negatively regulated by p53 [80]. Notably, androgen-insensitive subjects who lack functional ARs do not produce sebum and do not develop acne [81], underlining the important contribution of AR signaling in the pathogenesis of acne. 

## 3. Hypoxia-Inducible Factor-1α and Leptin

In 2014, Danby [82] postulated that ductal hypoxia may link comedogenesis and inflammation in acne. It has been hypothesized that ductal hypoxia as well as mTORC1 activation via the upregulation of HIF-1α may enhance the secretion of sebocyte-derived leptin into the ascending sebum, which may subsequently stimulate the proliferation of adjacent infundibular keratinocytes, promoting comedogenesis [83]. The key transcription factor of hypoxia is HIF-1α [84]. HIF-1α translation is upregulated by activated mTORC1 [47]. 

Remarkably, HIF-1α transactivates the human leptin (*LEP*) gene promoter [85]. It has been shown in human keratinocytes that leptin mediates mitogenic stimuli, promoting keratinocyte proliferation [86]. Under hypoxic conditions, primary keratinocytes induce filaggrin (*FLG*) gene expression in a HIF-1α- and HIF-2α-dependent manner [87]. In fact, increased filaggrin expression has been observed in the sebaceous ducts and infundibula of patients with acne vulgaris [88]. The upregulation of leptin-induced proinflammatory cytokines in normal human keratinocytes is mainly regulated via STAT3 signaling [89], which is also stimulated by overactivated mTORC1 [47]. 

HIF-1α acts via the retinoic acid receptor-related orphan receptor-γt (RORγt) to drive Th17 cell differentiation and is thus a key reprogrammer of metabolism in inflammatory cells promoting inflammatory gene expression [90]. HIF-1α controls the balance of Th17 cells and regulatory T cells [91]. Under hypoxic conditions, most eukaryotic cells can shift their primary metabolic strategy from predominantly mitochondrial respiration toward increased glycolysis to maintain ATP levels [92]. At the transcriptional level, this metabolic switch is critically dependent on HIF-1α, which induces the expression of glycolytic enzymes. Glycolysis generates indispensable metabolic intermediates [93] that are required for the rapid proliferation of keratinocytes [94,95], sebocytes [96] and Th17 cells [97], thus linking comedogenesis and Th17-cell-mediated inflammation in acne [82,98]. A pilot study indicated that the extraction of an acne lesion reduces HIF-1α expression [99]. Choi et al. [100] recently reported increased levels of SREBF1 and perilipin 2 (PLIN2) which were upregulated by HIF-1α in SZ95 sebocytes under hypoxia, indicating that a hypoxic microenvironment can increase lipogenesis and provide a link between seborrhea and inflammation. However, we must keep in mind that not only local hypoxia but also increased mTORC1 signaling generally enhance the expression of HIF-1α [47] and leptin [48], respectively. 

Surprisingly, there are no data on the effect of benzoyl peroxide, the most commonly used topical anti-acne agent, on ductal HIF-1α expression in skin with acne. Li et al. [101] reported that minocycline, an oral antibiotic commonly used for the treatment of inflammatory acne [102,103], induces the proteasomal degradation of HIF-1α under hypoxia by increasing the expression of prolyl hydroxylase-2 and the HIF-1α/von Hippel–Lindau protein interaction, thereby overcoming hypoxia-induced HIF-1α stabilization. Azelaic acid, another anti-acne agent used topically, might also interfere with HIF-1α because it is known to inhibit anaerobic glycolysis [104]. An intensive crosstalk between p53 and HIF-1α as mediators of molecular responses to physiological and genotoxic stresses has been reported [105]. Activated p53 decreases HIF-1α protein levels via accelerated proteasome-dependent degradation [106]. In contrast, AKT activation inhibits the p53-mediated degradation of HIF-1α [107]. Remarkably, PI3K-AKT activation is required for the hypoxic stabilization of HIF-1α, and hypoxia alone is not sufficient to render HIF-1α resistant to proteasomal cleavage and degradation [108]. Notably, we recently detected increased levels of expression of HIF-1α in the lesional skin of patients with acne inversa (hidradenitis suppurativa) [109].

## 4. Infundibular GATA-Binding Protein 6

Recent studies reported that the infundibulum and sebaceous ducts are lined by molecularly distinct differentiated cells which express markers including keratin 79 and the transcription factor GATA-binding protein 6 (GATA6) [110,111,112]. The loss of GATA6 causes the dilation of the hair follicle canal and sebaceous duct [111]. Importantly, Oulès et al. [112] observed reduced levels of expression of GATA6 in the upper pilosebaceous unit in acne patients. GATA6 controls the proliferation and differentiation of keratinocytes to prevent the hyperkeratinization of the infundibulum, which is the primary pathological event in acne vulgaris (comedogenesis). Okabe et al. [113] found the presence of retinoic acid response elements (RAREs) in the putative regulatory region of the *GATA6* gene and identified retinoic acid as a critical signal inducing GATA6 transcription. The topical treatment of acne with *all-trans* retinoic acid (ATRA) and its systemic treatment with isotretinoin, the precursor of ATRA, may thus normalize disturbed follicular keratinization, promoting comedolysis via the induction of GATA6 expression. Importantly, the expression of GATA6 is also upregulated by FoxO1 [112]. *GATA6* exhibits three putative FoxO1 binding sites. The inhibition of AKT reduces phosphorylated FoxO1 and increases the nuclear localization of FoxO1, subsequently enhancing GATA6 expression [114]. In contrast, the overexpression of IGF-1/AKT signaling may thus reduce infundibular *GATA6* expression, promoting comedogenesis via the suppression of nuclear FoxO1 and GATA6. mTORC2 signaling, the androgen-driven pathway to activate AKT, also negatively regulates *GATA6* expression in a FoxO1-dependent manner [114]. Thus, increased insulin/IGF-1-AKT signaling and androgen-mTORC2-AKT signaling converge in the suppression of GATA6. Intriguingly, the overexpression of GATA6 suppresses AR expression [110], pointing to enhanced infundibular AR signaling in reduced states of GATA6. Notably, the inhibition of mTORC1 stabilizes GATA6 and promotes the nuclear accumulation of GATA6 [115]. In accordance with Oulès et al. [112], GATA6 expression contributes to the therapeutic effect of ATRA, the main topical treatment for comedonal acne. The isotretinoin/ATRA-induced overexpression of p53 and FoxO1 with the associated attenuation of PI3K-AKT-mTORC1 signaling may stabilize infundibular GATA6 in acne, a potential key mechanism of comedolysis. The treatment of human SEB-1 sebocytes with isotretinoin induces cell cycle arrest, which is associated with the increased expression of p21 (cyclin-dependent kinase inhibitor 1A; *CDKN1A*) [116]. The *CDKN1A* gene is transcriptionally activated by p53 [117]. Remarkably, transient transfection with a GATA6 expression vector inhibited S-phase entry in vascular smooth muscle cells and in mouse embryonic fibroblasts lacking both p53 alleles. The GATA6-induced growth arrest correlated with a marked increase in the expression of p21 [118]. Thus, there appear to be close interactions between the isotretinoin-induced upregulation of p53-FoxO1, p53-p21 and FoxO1-GATA6. In contrast, hypoxia-induced microRNA-181b has been shown to target *GATA6* [119], thus eventually destabilizing GATA6-controlled infundibular keratinocyte homeostasis, linking ductal hypoxia to ductal GATA6 deficiency in acne.

## 5. Transforming Growth Factor β

Genome-wide association studies identified gene loci associated with impaired transforming growth factor β (TGFβ) signaling, including the genes *TGFB2*, *OVOL1* and *FST* [120,121]. In fact, experimental evidence supports the view that TGFβ signaling is reduced in the SGs of acne patients [112]. The activation of the TGFβ signaling pathway is necessary and sufficient for maintaining sebocytes in an undifferentiated state. The presence of TGFβ suppresses genes required for the production of sebaceous lipids and for sebocyte differentiation, such as fatty acid desaturase 2 (FADS2) and PPARγ, thereby decreasing lipid accumulation through a TGFβR2-SMAD2-dependent pathway [122]. There is an important molecular crosstalk between GATA6 and TGFβ. GATA6 is a critical signal for the activation of TGFβ [113]. In a sebaceous organoid model, Oulès et al. [112] obtained evidence that GATA6-mediated TGFβ activation is a key process controlling the repression of the interfollicular fate of keratinocytes and promoting junctional zone/sebaceous duct differentiation and the downregulation of the infundibular differentiation program [112]. Furthermore, it has been shown in chondrocytes that TGFβ stimulates FoxO1 expression [123] pointing to crosstalk between GATA6, TGFβ and FoxO1.

## 6. Cutibacterium Acnes

Obstruction of the pilosebaceous unit leads to hypoxia, favoring the development of *Cutibacterium acnes* (*C. acnes*), formerly designated *Propionibacterium acnes* (*P. acnes*). Multiple studies suggest that dysbiosis of the skin microbiota significantly contributes to the development of acne. *C. acnes* is the dominant resident bacterial species in SG-enriched areas of the skin implicated in the pathogenesis of acne and disease progression [124]. Acne has been linked to dysregulated innate immunity in response to *C. acnes* promoting increased Th17 cell differentiation with increased IL-17 levels [125], thus contributing to the sebocyte-mediated polarization of cutaneous T cells toward the Th17 phenotype [126]. To mimic the effect of *C. acnes* on sebocytes, Oulès et al. [112] treated control or GATA6-expressing sebocytes with peptidoglycan (PGN), which is the main component of the Gram+ bacterial cell wall, which is recognized by toll-like receptor 2 (TLR2). GATA6 expression led to a decrease in PGN-induced IL-17 expression. When they explored the effect of live *C. acnes* bacteria on control and GATA6-expressing sebocytes, GATA6 expression showed a trend toward deceased IL-17 expression [112]. It has been reported that *C. acnes* membrane fractions increase IGF-1 and IGF1R expression in the epidermis of explants [127], a constellation that may reduce the FoxO1-mediated expression of GATA6. *C. acnes* also induces TLR2 and TLR4 on keratinocytes, a mechanism that could play an essential role in acne-linked inflammation [128] as TLR2 activation drives an inflammatory transcriptional program in Th17 cells [129]. *C. acnes* biofilms, in contrast to planktonic bacteria, are characterized by upregulated stress-induced genes and the upregulation of genes coding for potential virulence-associated CAMP factors [130], including exogeneous triacylglycerol lipase (*gehA*) [131], enhancing the release of free fatty acids like palmitic and oleic acid [132,133]. Lipase-mediated triacylglycerol hydrolysis thus generates saturated free fatty acids that bind and activate TLR2-triggered proinflammatory signaling [134]. The gene expression levels of interleukin 8 (IL-8) and TLR2 were enhanced by cell-free extracts of *C. acnes* in SZ95 sebocytes [135]. Palmitic acid can stimulate the production of IL-6, TNF-α and IL-1β in HaCaT keratinocytes promoting cell proliferation, potentially contributing to inflammation and the hyperkeratinization of the pilosebaceous duct in acne [136]. 

Taken together, *C. acnes*-derived signaling augments a variety of transcriptomic deviations observed in acne vulgaris.

## 7. Transcriptomic Effects of Isotretinoin Treatment 

### 7.1. Isotretinoin’s In Vitro versus In Vivo Gene-Regulatory Effects

In contrast to ATRA, isotretinoin has little or no ability to bind to cellular retinol-binding proteins, nuclear retinoic acid receptors (RARs) or retinoid X receptors (RXRs) but may act as a prodrug that is converted intracellularly into metabolites that are agonists for RARs and RXRs [137,138,139,140]. At least five major metabolites of isotretinoin exist: ATRA, *13-cis*-4-oxo-retinoic acid, *all-trans*-4-oxo-retinoic acid, *9-cis*-retinoic acid and *9-cis*-4-oxo-retinoic acid [141,142,143,144,145,146]. The incubation of SZ95 human sebocytes with isotretinoin resulted in significantly higher intracellular concentrations of ATRA than isotretinoin [147]. Their incubation with ATRA generated very high intracellular concentrations of ATRA and negligible concentrations of isotretinoin, suggesting that ATRA may be the active intracellular form of isotretinoin; this prompted Tsukada et al. [147] to conclude that isotretinoin should be considered a prodrug. 

Contradictorily, the apoptosis observed in immortalized SEB-1 sebocytes after exposure to isotretinoin over a short period (48 h, 72 h) could not be recapitulated by ATRA and could not be inhibited by the presence of an RAR pan-antagonist, which led the investigators to conclude that isotretinoin induces apoptosis as well as sebum suppression via an RAR-independent mechanism [116]. However, these in vitro observations after short-term isotretinoin exposure do not represent the histologically and clinically observed time course of the involution of the SGs in isotretinoin-treated patients, which range from 49% SG involution at 1 week to 76% after 8 weeks [148]. Gene profiling studies of immortalized SZ95 sebocytes treated short-term (6 h and 24 h) with isotretinoin did not find upregulated levels of gene expression for *CRABP2*, *FOXO1*, *FOXO3* or *TP53* [149]. 

In contrast, a study of temporal changes in gene expression in the skin of isotretinoin-treated patients noted increased expression of *CRABP2* after one week of treatment [148]. In accordance, increased *CRABP2* expression was previously demonstrated in the patients’ SGs following 3–16 weeks of isotretinoin therapy [150]. Notably, the human *CRABP2* gene is regulated by a far-upstream RARE that most efficiently binds RAR-RXR heterodimers [151,152,153]. Hence, CRABP2 is regarded as a useful marker of isotretinoin action in the SGs of acne patients [150]. The selective induction of suprabasal CRABP2 in SGs indicates that ATRA is the preferential isotretinoin derivative for isotretinoin’s mode of action [150]. It is of critical functional importance that the partitioning of ATRA between the nuclear receptors RAR and PPARβ/δ is regulated by CRABP2 and fatty-acid-binding protein 5 (FABP5) [154]. In cells with a high CRABP2/FABP5 ratio, ATRA functions through RARs and is a proapoptotic agent, but in cells that highly express FABP5, it signals through PPARβ/δ and promotes cell survival [154]. TUNEL staining (an indicator of apoptosis) was the strongest in the nuclei of sebocytes in the basal layer of the SG and in early differentiated sebocytes adjacent to the basal layer of patients after 1 week of isotretinoin treatment [155]. It is thus conceivable that sustained isotretinoin exposure enforces suprabasal CRAPB2-mediated ATRA/RAR-signaling, which may upregulate the expression of p53 as a second retinoid-induced transcriptional response [156]. In a third wave of transcriptomic changes, overexpressed p53 may finally orchestrate cell cycle arrest, autophagy and apoptosis, resulting in sebum suppression (Figure 2).

Taken together, short-term (hours/days) in vitro sebocyte cultures (of single cells) may not be suitable for providing a realistic in vivo picture of observed long-term (weeks) isotretinoin signaling in the sebaceous glands (cells of a differentiating tissue) of acne patients exhibiting basal/suprabasal CRABP2 expression and apoptosis. 

### 7.2. Isotretinoin’s Potential Impact on Sebocyte Stem and Progenitor Cells

There is recent interest in the role of stem and progenitor cells in controlling SG development, physiology, pathology and aging [157,158,159]. Recently, Veniaminova et al. [159] reported that SGs are largely self-renewed by dedicated stem cell pools during homeostasis. Retinoids play a key role in the regulation of stem cell differentiation [156,160,161]. B lymphocyte-induced maturation protein 1 (BLIMP1), encoded on the *PRDM1* gene, defines a progenitor population that governs cellular input to the SG [162]. In fact, single Blimp1(+) cells isolated from mice have the potential to generate SG organoids in vitro [163]. Intriguingly, mice with a chronic activation of p53 develop an aging phenotype in the skin associated with a reduction in subcutaneous fat and the loss of SGs [164]. In these mice, Blimp1(+) SG progenitor cells became depleted, resulting in the atrophy of the entire SG [164]. Furthermore, the activation of p53 can inhibit c-MYC-induced SG differentiation [165], whereas the overexpression of c-MYC stimulates sebocyte differentiation [166]. BLIMP1 binds to the *TP53* promoter and represses p53 transcription. The suppression of p53 transcription is a crucial function of endogenous BLIMP1 and is essential for normal cell growth [167]. Recent evidence indicates that FoxO1 is a negative regulator of BLIMP1 expression [168].

Persistent isotretinoin-mediated upregulation of p53 and FoxO1 may thus deplete the BLIMP1(+) progenitor cell pool, promoting SG involution.

### 7.3. P53/FoxO1-Mediated Suppression of IGF-1/IGF1R/PI3K/AKT/mTORC1 Signaling

Oral isotretinoin is the most effective treatment for recalcitrant and severe acne, improving all major aspects of the pathogenesis of acne [169,170]. It was predicted that isotretinoin upregulates the expression of FoxO1 and FoxO3 [171,172,173]. At present, experimental evidence has confirmed that isotretinoin enhances the expression of FoxO1 in primary human keratinocytes [174], immortalized SZ95 human sebocytes [175] and the SGs of isotretinoin-treated acne patients [176]. We demonstrated via the immunohistochemistry of SGs that isotretinoin treatment increased the nuclear accumulation of FoxO1 and FoxO3 proteins [176], pointing to changes in FoxO-mediated gene expression or FoxO translocation. Furthermore, it has been hypothesized that isotretinoin upregulates p53 in the pilosebaceous unit [35]. Notably, p53 maintains the baseline expression of *FOXO1A* [177]. Importantly, *FOXO3A* has also been confirmed to be target gene of p53 [178]. Experimental evidence confirmed that the treatment of primary human keratinocytes with isotretinoin increased the expression of p53 [174]. Recently, we demonstrated that 6 weeks of oral isotretinoin treatment enhanced the nuclear accumulation of p53 in the SGs of acne patients [179]. Notably, prior to the isotretinoin treatment, the acne patients exhibited lower cutaneous expression levels of the p53 protein compared to acne-free controls [179]. 

p53, FoxO1 and FoxO3 are fundamental negative regulators of cell cycle progression and cell proliferation. These proapoptotic transcription factors exert synergistic effects in promoting catabolism, autophagy and apoptosis, respectively [180,181,182,183,184,185,186] (Figure 2).

p53 suppresses IGF-1/PI3K/AKT/mTORC1 activity at multiple regulatory checkpoints [187,188,189,190,191,192,193,194,195,196]. The *IGF1* promoter is negatively regulated by p53 [190]. In fact, Karadag et al. [192] observed decreased serum levels of IGF-1 in isotretinoin-treated acne patients. In addition, *IGF1R* expression is also negatively regulated by p53 [193]. The disruption of endogenous IGF1R led to the inhibition of insulin receptor (*INSR*) promoter activity by p53 [194]. Notably, the expression levels of negative regulators of the PI3K-AKT-mTORC1 signaling pathway, including phosphatase and tensin homolog (*PTEN*) (a negative regulator of PI3K), *TSC2* (a negative regulator of RHEB) and AMP-activated protein kinase β1 (*PRKAB1*) (an activator of TSC2), are all upregulated by p53 [195]. DEP domain-containing mTOR-interacting protein (DEPTOR) is a natural inhibitor of mTORC1 and mTORC2. *DEPTOR* is also a downstream target of p53, whose activity positively correlates with DEPTOR expression [197].

Taken together, the isotretinoin-mediated upregulation of p53 counteracts multiple regulatory checkpoints of exaggerated IGF-1/IGF1R/PI3K/AKT-mTORC1 signaling pathways and potentially mTORC2 signaling pathways in acne patients (Figure 3).

### 7.4. P53/FoxO1 Upregulation Suppresses AR Signaling

In a murine model, Cottle et al. [165] demonstrated that p53 activation inhibits SG differentiation and disrupts AR signaling. It has been shown that the upregulation of p53 inhibits AR expression [80,198]. In fact, oral isotretinoin treatment reduced AR levels in the skin of male acne patients [199]. 

Thus, isotretinoin attenuates IGF-1/IGF1R and AR signaling, two critical converging pathways involved in AKT/mTORC1-driven acne pathogenesis [200] (Figure 3). 

### 7.5. GATA6 Upregulation Suppresses Comedogenesis

Reduced infundibular *GATA6* expression has been related to disturbed follicular keratinization (comedo formation) in acne patients [112]. Systemic isotretinoin treatment has a two principal impacts on the upregulation of infundibular *GATA6* expression: (1) After the isomerization of isotretinoin (*13-cis* retinoic acid) into ATRA [147], ATRA activates RARs [201], enhancing GATA6 expression via binding RARs to RAREs on *GATA6* promoters [113]. (2) In addition, the p53-mediated upregulation of nuclear FoxO1 may activate three FoxO1 binding sites on the *GATA6* promoter [112,114], further promoting its expression. Upregulated GATA6 represses AR activation [110], a synergistic interplay normalizing disturbed infundibular keratinization (comedogenesis) in acne patients.

### 7.6. Perilipin 2-Mediated Suppression of Comedogenesis

The proteome of lipid droplets (LDs) has emerged as a major influencer of various aspects of LD biology [202]. Perilipins (PLINs) are the most studied and abundant proteins residing on the LD’s surface [202]. Recently, LD protein expression has been characterized in SGs, highlighting their roles in sebocyte physiology, sebaceous lipogenesis and the pathogenesis of acne [203,204,205]. PLIN2 is the major perilipin involved in sebocyte differentiation and controlling sebaceous lipid accumulation and SG size [205]. One general function of PLINs is to restrict the access of lipases to LDs, thus preventing lipolysis [206]. This gatekeeper effect on lipolysis has also been reported for PLIN2 [207,208]. Sorg et al. [209] recently observed reduced infundibular expression of PLIN2 in acne patients compared with acne-free controls. In contrast, treatment with *Silybum marianum* fruit extract resulted in increased PLIN2 levels that correlated with lower ongoing comedogenesis. The release of free fatty acids from sebum triacylglycerols was significantly decreased, supporting the role of PLIN2 in the prevention of the release of comedogenic free fatty acids [209].

Interestingly, silibinin, the active constituent extracted from *Silybum marianum*, has been shown to activate p53 [210,211,212], pointing to a potential contribution of p53 in the silibinin-induced upregulation of sebocyte PLIN2. In accordance, it has been shown in human skeletal muscle cells that PLIN2 expression strongly correlates with increased p53 activation and reduced IGF-1 expression [213,214]. 

Thus, the isotretinoin-mediated upregulation of p53 may enhance sebocyte PLIN2 expression, preventing sebaceous triacylglycerol hydrolysis and reducing the release of comedogenic free fatty acids.

### 7.7. Sebum Suppression, Sebocyte Autophagy and Apoptosis

It is well accepted that the binding of ATRA initiates changes in the interactions of RAR/RXRs with co-repressor and co-activator proteins, activating the transcription of primary target genes, alters interactions with proteins that induce epigenetic changes and induces transcription of genes encoding transcription factors and signaling proteins that further modify gene expression [156]. The isotretinoin-induced upregulation of p53 explains sebocyte apoptosis and sebum suppression as the major desired pharmacological effects in the treatment of acne [35,215]. Nelson et al. [216] demonstrated that tumor necrosis factor-related apoptosis-inducing ligand (TRAIL) contributes to the apoptotic effect of isotretinoin in human sebocytes. Importantly, p53 upregulates the expression of the proapoptotic proteins TRAIL (*TNFSF10*) [217], FoxO1 (*FOXO1A*) [176,177] and FoxO3 (*FOXO3A*) [176], respectively. In contrast, survivin (*BIRC5*), a member of the inhibitors of the apoptosis gene family, which is overexpressed in the serum of acne patients [218,219], is suppressed by p53 [220]. Thus, the isotretinoin-mediated upregulation of nuclear p53 in SGs [179] most likely represents the desired sebum-suppressive effect in promoting sebocyte apoptosis. In SGs, isotretinoin (*13-cis* retinoic acid) is converted into *all-trans* retinoic acid (ATRA) [147]. Via binding to cellular retinoic-acid-binding protein 2 (CRABP2), ATRA is transported into the nucleus, where the transcription of p53 is upregulated in an RAR-dependent fashion [221]. The ATRA/RAR-mediated upregulation of the expression of p53 was also reported in other benign and malignant cells [222,223,224,225,226,227,228]. Preferential CRABP2 expression in the suprabasal layers of SGs is mandatory for ATRA-induced transcriptomic changes, resulting in sufficient sebum suppression [150]. It is of critical functional importance that the partitioning of ATRA between the nuclear receptors RAR and PPARβ/δ is regulated by the intracellular-lipid binding proteins CRABP2 and fatty-acid-binding protein 5 (FABP5) [154]. The increased expression of CRABP2 in the isotretinoin-treated SGs of patients with acne was observed after weeks of oral isotretinoin exposure [150], whereas following short-term isotretinoin exposure, immortalized sebocytes neither exhibited increased CRABP2 nor upregulated p53, FoxO1 or FoxO3 expression [148,149,229]. 

Recently, Seo et al. [175] reported that autophagy is constitutively active in the maturing sebocytes of human SGs, whereas autophagy-related protein expression is repressed in the SGs of acne patients. The authors found that isotretinoin activates autophagy in immortalized SZ95 sebocytes and induces sebosuppression [175]. Blocking autophagy using siRNA targeting autophagy-related 7 (ATG7) resulted in a significant loss of the sebosuppressive activity of isotretinoin in SZ95 sebocytes stimulated using testosterone and linoleic acid. Unfortunately, the investigators overlooked the reported intimate interaction between ATG7 and p53. ATG7 binding to p53 enhances the transcription of the cell-cycle inhibitor p21 [230]. Among the genes directly activated by p53 are several autophagy genes, including ATG7 [231]. It is well established that there is an important relationship between autophagy and p53. Autophagy suppresses p53, whereas p53 activates autophagy [232]. In this regard, studying autophagy in immortalized SZ95 sebocytes is critical because the simian virus (SV) transfection used for sebocyte immortalization is mediated via the inactivation of p53 by the direct binding of SV40 large T antigen to p53 [233,234]. Proapoptotic p53 signaling may thus be compromised in p53-inactivated immortalized sebocytes, which may thus be unable to execute a complete program of p53-induced apoptosis and instead remain in a preliminary state of autophagy. In contrast, human sebocytes in vivo are not artificially p53-inactivated, and the clinical period to obtain a histological “involution” of SGs in acne patients takes weeks of isotretinoin treatment compared to short-term cell-culture studies [235].

### 7.8. Teratogenicity and Neural Crest Cell Apoptosis

Neural crest cells (NCCs) and neural crest (NC)-derived neuroblastoma cells are very susceptible to isotretinoin-induced apoptosis [215,236,237]. Neural precursor cells possess multiple p53-dependent apoptotic pathways [238]. Translational evidence suggests that the isotretinoin-induced upregulation of p53 promotes NCC apoptosis, which is suggested to operate as the major pathogenic mechanism of isotretinoin’s teratogenicity [239]. Increased p53 signaling is also associated with Treacher Collins, CHARGE and fetal alcohol syndrome, which all exhibit dysmorphic craniofacial features resembling retinoid embryopathy [239]. It was demonstrated in embryonic mouse fibroblasts that p53 controls the NC/EMT gene network [240]. The upregulation or stabilization of p53 in the cranial neural tube reduces cranial NC delamination and promotes neural tube defects in chick embryos [240]. p53 was shown to play a major role in Treacher Collins syndrome, a congenital haploinsufficiency disorder in humans that arises from mutations in the *TCOF1* gene. In the absence of one *Tcof1* allele in mice, the upregulation of p53-related apoptotic genes in NC progenitors results in severe craniofacial defects [241]. Keeping in mind that the pathogenic mechanisms involved in Treacher Collins syndrome are dependent on the p53 pathway, results from a recent in vivo ATRA treatment showed similar processes in which ATRA exposure led to an increase in apoptosis processes at late time points and could thus participate in producing developmental birth defects [242]. Using a proteomics approach, a new ATRA target, *EFTUD2* (elongation factor Tu GTP-binding domain-containing 2) was identified whose dysregulation leads to craniofacial defects [242]. The homozygous deletion of *Eftud2* causes brain and craniofacial malformations, affecting the same precursors as in mandibulofacial dysostosis with microcephaly patients [243]. Remarkably, increased p53 activity and NCC death are responsible for craniofacial malformations in the *Eftud2; Wnt1-Cre2* mutant mouse model [243]. The overactivation of the p53 pathway in *Eftud2* knockdown cells was attenuated via the overexpression of non-spliced MDM2, and craniofacial development was improved when *Eftud2*-mutant embryos were treated using pifithrin-α, an inhibitor of p53 [243].

It is important to note that Blimp-1 specifies NC and sensory neuron progenitors in the zebrafish embryo [244]. The p53- and FoxO1-mediated suppression of BLIMP1 may thus explain isotretinoin’s teratogenic activity at the level of NC progenitor cells. Thus, accumulating evidence underlines the role of overactivated p53/FoxO1 signaling in NCC apoptosis, linking syndromes with dysmorphic craniofacial features with the pathogenesis of isotretinoin-induced embryopathy (teratogenicity).

### 7.9. Depression and Impaired Hippocampal Neurogenesis

The question as to whether isotretinoin causes depression and anxiety in acne patients is still a matter of debate [245,246,247,248,249,250]. However, a subgroup of patients may be at risk of developing depression or suicidal ideation when undergoing isotretinoin treatment [246,249]. The hippocampus is one of the brain regions in which new neurons are constantly formed in a process called hippocampal neurogenesis [251,252,253,254]. Reduced hippocampal volume and low numbers of hippocampal neural progenitors have been reported in depressed humans [254]. The commonly used anti-depressive drug lithium has been shown to increase hippocampal neurogenesis in rodents and humans [255,256,257,258]. Long-term lithium therapy has been shown to suppress neuronal p53 levels [259,260] and reduces the transcriptional activity of FoxO3 by decreasing its intracellular content [261]. Intriguingly, acne is a possible adverse effect of lithium therapy [262,263,264], a potential result of attenuated p53/FoxO signaling. Remarkably, the treatment of mice with isotretinoin results in both decreased hippocampal neurogenesis and a reduction in hippocampal volume [265,266]. The treatment of hypothalamic cells with 10 μM of isotretinoin for 48 h decreased cell growth to 45.6 ± 13% compared to a control [266]. Griffin et al. [266] hypothesized that the ability of isotretinoin to decrease hypothalamic cell numbers may contribute to the increase in depression-related behaviors observed in mice. ATRA applied intracerebroventricularly to adult rats increased RARα protein expression in the hippocampus, suggesting an activation of ATRA/RARα-induced signaling mechanisms [267]. In these rats, ATRA-induced impairments in hippocampal neurogenesis correlated with depression-like symptoms [267]. Remarkably, retinoic acid-inducible gene 1 (*RAI1*) was found to be significantly upregulated in brains from patients with schizophrenia, bipolar disorder or major depression [268]. Gene expression profiling revealed the role of RAI1-like receptor signaling in p53-dependent apoptosis induced via psoralen + UV-A (PUVA) in keratinocytes [269]. A direct relationship between p53 expression and the loss of viability in CNS neurons [270] and neuronal cell death is well established [271,272,273]. In fact, the adenovirus-mediated delivery of the p53 gene causes cortical and hippocampal neuronal cell death, with some features typical of apoptosis [274]. 

Recent studies have shown that FoxOs are also implicated in the pathophysiology of depression [275,276]. They play an essential role in neural stem cell homeostasis [277]. In gerbil and mouse brains the dephosphorylation of FoxO1 following transient forebrain ischemia resulted in the translocation of FoxO1 into the nucleus in neurons [278]. The activation of FoxO1 preceded delayed neuronal death in the vulnerable hippocampal regions following ischemic brain injury. Notably, FoxO1 activation was accompanied by an increase in DNA binding activity for FoxO1-responsive element on the Fas ligand promoter (*FASLG*) [278]. In accordance with the AKT-mediated regulation of FoxO1, the AKT-induced phosphorylation of FoxO3 promotes cell survival via the extrusion of FoxO3 from the nucleus into the cytoplasm [279,280], whereas nuclear non-phosphorylated FoxO3 triggers apoptosis, inducing the expression of *FASLG* [280]. Notably, *FOXO1A* is a target of FoxO3, which induces the expression of FoxO1 [281]. The IGF-1-induced AKT-mediated phosphorylation of FoxO3 has been observed in NC-derived PC12 cells in which IGF-1 inhibits the apoptosis of neuronal cells [282]. Studies in murine models confirmed that FoxO1 and FoxO3 influence behavioral processes linked to anxiety and depression [283]. In fact, the IGF-1-PI3K-AKT-mediated phosphorylation of FoxO3 induces the survival of cultured hippocampal neurons [284]. In contrast, the overexpression of FoxO3 induces the apoptosis of cultured hippocampal neurons [284]. In this regard, it is of critical concern that we observed the nuclear overexpression of both FoxO1 and FoxO3 in the SGs of acne patients treated with isotretinoin [176], a pharmacological constellation that may negatively affect the homeostasis of hypothalamic and hippocampal neurons. In accordance, Kim et al. [285] showed that the overexpression of FoxO1 suppresses neuronal differentiation, whereas in the absence of FoxO-dependent homeostatic processes, there was a significant decrease in the neuronal stem cell pool and accompanying neurogenesis in the adult mouse brain. 

Thus, the isotretinoin-mediated upregulation of p53, FoxO1 and FoxO3 might negatively affect hippocampal neurogenesis, eventually increasing the risk of depression in predisposed individuals.

### 7.10. Reduced Ovarian Reserve and Granulosa Cell Apoptosis

Systemic isotretinoin also modifies the pituitary–ovarian axis, causing a mild suppression of pituitary hormone levels including growth hormone (GH), thyroid stimulating hormone (TSH), prolactin, adrenocorticotropic hormone (ACTH) and luteinizing hormone (LH) [286,287]. Follicle-stimulating hormone (FSH) and LH are required for the maturation of ovarian follicles. FoxO1 is critically involved in the regulation of gonadotropin expression [288,289,290,291]. Increased nuclear localization of FoxO1 decreases FSH β-polypeptide (*FSHB*) mRNA levels in murine primary pituitary cells [289]. FoxO1 overexpression in pituitary gonadotrope cells also inhibits the transcription of the β-subunit of LH (*LHB*) [290]. 

In rats, isotretinoin reduced ovarian reserve, a process associated with granulosa cell (GC) apoptosis [292]. In women treated with isotretinoin, the anti-Müllerian hormone level, the antral follicle count and ovarian volume were significantly reduced compared to pretreatment values, pointing to negative effects on ovarian reserve [293]. FoxO1 and FoxO3 are highly expressed in the GCs of ovarian follicles. They interact with activin to regulate genes controlling follicle growth or with bone morphogenic protein 2 (BMP2) to control genes associated with metabolic stress and apoptosis leading to follicle death [294]. The overexpression of FoxO1 inhibits the viability of GCs in mice [295]. The constitutive activation of FoxO1 in murine GCs not only abolishes the protection from FSH but activates autophagic gene expression [296]. Further experimental evidence in mice and chickens supports critical roles of FoxO1 and FoxO3 in the apoptosis of GCs [297,298]. In addition, FSH and FoxO1 regulate genes in the sterol/steroid and lipid biosynthetic pathways in GCs [299]. It was shown in porcine GCs that p53 promotes apoptosis and suppresses cell proliferation [300]. 

The impact of isotretinoin action on the pituitary–ovarian axis and GCs might also be caused by overexpressed p53/FoxO1/FoxO3 signaling, resulting in gonadotropin suppression and GC apoptosis, respectively [301].

### 7.11. Hypertriglyceridemia

The isotretinoin-mediated upregulation of p53 and FoxO1 also provides a reasonable explanation for isotretinoin-induced hyperlipidemia, which results primarily from the hepatic oversecretion of apolipoprotein B100-containing very-low-density lipoproteins (VLDLs) [302,303]. Gustafson et al. [304] showed that rats respond with a prompt increase (+ 250%) in plasma triacylglycerol levels within the first few days after starting treatment with isotretinoin, which was exclusively due to a marked increase in VLDLs. We observed a significant increase in VLDL apolipoprotein B (apoB) levels during the treatment of acne patients with isotretinoin [303]. The hepatic synthesis of apo B (B-100) is absolutely required for the assembly and secretion of VLDLs. It contains several very hydrophobic areas that serve as strong lipid-binding domains [305,306]. Intriguingly, the *APOB* gene has been identified as a target gene of p53 [307]. The loading of hepatic triglycerides onto apoB100 is mediated by microsomal triglyceride transfer protein (MTTP), which is also essential for the assembly and secretion of VLDLs and is induced by FoxO1 [308]. In addition, FoxO1 stimulates hepatic apolipoprotein C-III (*APOC3*) expression. FoxO1 binds to its consensus site in the *APOC3* promoter [309,310]. In men, isotretinoin treatment (80 mg/d; 5 d) resulted in elevated plasma apoC-III concentrations at the transcriptional levels [311]. It has been shown that apoC-III strongly inhibits the hepatic uptake of VLDLs and intermediate-density lipoproteins (IDLs) [312] and inhibits the activity of lipoprotein lipase [313,314]. 

Thus, isotretinoin-mediated changes with the transcriptional upregulation of p53 and FoxO1 explain isotretinoin-induced hypertriglyceridemia via increased hepatic VLDL apoB100 synthesis, triglyceride loading onto VLDLs and hepatic VLDL secretion as well as impaired VLDL triglyceride hydrolysis and hepatic VLDL and IDL uptake. 

### 7.12. Increased Transepidermal Water Loss and Dry Skin

Dry skin is the most common mucocutaneous side effect of oral isotretinoin treatment [315]. Increased transepidermal water loss induced by oral isotretinoin treatment has been related to the upregulated expression of aquaporin 3 (*AQP3*) [316]. AQP3 controls a water, glycerol and hydrogen-peroxide-transporting channel that plays a key role in various processes involved in keratinocyte function [317]. *AQP3* is another p53 target gene [318]. 

Isotretinoin-induced p53 overexpression in epidermal keratinocytes [174] thus explains increased AQP3-mediated transepidermal water loss resulting in dry skin.

### 7.13. Intracranial Hypertension

Intracranial hypertension (*pseudotumor cerebri*) is a potential adverse effect of oral isotretinoin treatment [319,320]. An association between increased choroid plexus (ChP) aquaporin 1 (*AQP1*) and intracranial pressure has also been observed in obese Zucker rats [321]. AQP1 plays an essential role in the movement of water through ChP epithelial cells. Notably, CRABP2 is specifically expressed in ChP [322]. AQP1 is primarily located in the apical membrane of the ChP epithelium and has been implicated in playing a pivotal role in cerebrospinal fluid secretion [323]. AQP1 is widely distributed in the human brain and is associated with the secretion of water into the subarachnoid space and has been suggested to play a key role in idiopathic and drug-induced intracranial hypertension [324]. A close correlation between the upregulated expression of p53, p21 and AQP1 has been reported in the rat kidney, heart, lung and small intestine [325]. In fact, a recent bioinformatic analysis of the *AQP1* promoter revealed the presence of DNA binding sites for p53 [326]. 

Thus, the isotretinoin-induced overexpression of p53 in ChP epithelial cells may explain the occurrence of intracranial hypertension in predisposed individuals.

### 7.14. Inflammatory Flare upon the Initiation of Isotretinoin Treatment

Interleukin 1β (*IL1B*) mRNA and the active, processed form of IL-1β are abundant in inflammatory acne lesions [327,328,329]. Caspase 1, also known as IL-1 converting enzyme, is a protease responsible for the processing of the key proinflammatory cytokine IL-1β from the inactive precursor pro-IL-1β into the active, secreted IL-1β [330,331]. In addition to the processing of IL-1β, caspase-1 plays an important and a conserved role as a cell death protease [329]. Remarkably, *CASP1* is a target gene of p53 [332]. Thus, the initiation of isotretinoin therapy with the upregulation of p53 might explain the inflammatory acne flare at the beginning of isotretinoin treatment [333] and the occurrence of acne fulminans after the initiation of isotretinoin therapy [334,335]. Growing evidence suggests that many of the signaling molecules known to regulate programmed cell death can also modulate inflammasome activation in a cell-intrinsic manner [336]. 

## 8. Limitations of Immortalized Human Sebocytes

In 1979, p53 was first identified in complex with the SV40 tumor-virus oncoprotein [337,338,339,340]. The immortalized human sebocyte cell lines SZ95 [341] and SEB-1 [342] were generated in 1999 and 2001, respectively. They are widely used tools for acne research and the exploration of pharmaceutical effects of potential anti-acne agents. Both immortalized sebocyte cell lines are generated via simian virus (SV) transfection [341,342]. Importantly, SV40 large T antigen binds and inactivates p53 [233,234]. Furthermore, it has been shown that SV40 T antigen inhibits binding of p53 to cellular DNA [343]. As p53, which is called the guardian of the genome [344], plays key roles in the regulation of lipid metabolism [345,346,347,348,349,350], cell cycle control [351,352], autophagy [232] and apoptosis [351,352,353,354,355], the viral inactivation of p53 via SV40 transfection will certainly cause deviations from normal physiological and pharmacological cellular responses compared to non-immortalized primary human sebocytes or in vivo SG specimens obtained from human subjects in whom p53 is not artificially inactivated. Although an immortalized cell should generally be resistant against cell death, Wróbel et al. [356] reported apoptosis in human immortalized SZ95 sebocytes after 6 h of SZ95 sebocyte treatment with high concentrations of staurosporine (10^−6^–10^−5^ M). However, treatment with *13-cis* retinoic acid (10^−8^–10^−5^ M) did not affect externalized phosphatidylserine levels, DNA fragmentation or lactate dehydrogenase cell release despite increased caspase 3 levels, leading to the conclusion that isotretinoin (*13-cis* retinoic acid) did not execute programmed death in human SZ95 sebocytes [356]. 

From the current perspective, these former experiments require a critical re-evaluation: p53-inactivated SZ95 sebocytes apparently fail to induce sufficient isotretinoin-mediated upregulation of p53, resulting in the incomplete execution of apoptosis. In addition, immortalized sebocytes treated with isotretinoin exhibit peculiar aberrations in the regulation of lipid metabolism. Human SEB-1 sebocytes subjected to short-term exposure to *13-cis* retinoic acid (10^−7^ M, 10^−6^ M and 10^−5^ M) for 48 h did not exhibit apoptosis but paradoxically, dose-dependent increases in total lipid synthesis and SREBF1 and 5-lipoxigenase (ALOX5) expression were induced [229]. Of note, it has been reported that SV40 large T antigen/p53 complex activates the promoter of the *IGF1* gene [357]. The isotretinoin-induced upregulation of p53 in SEB-1 cells may thus have increased the formation of SV40 large T antigen/p53 complex, enhancing IGF-1-AKT-SREBF1 signaling and enhancing lipogenesis [358]. A partial increase in the isotretinoin-mediated upregulation of p53 in SEB-1 sebocytes may also explain the observed upregulation of *ALOX5*, a further target gene of p53 [359]. *SREBF1* expression is, under physiological conditions, negatively regulated by p53 [350], pointing to insufficient isotretinoin/p53-mediated SREBF1 suppression in immortalized SEB-1 sebocytes. Also, the statement that “isotretinoin is indirectly effective in sebocytes” [149] requires a re-evaluation. The metabolic effects observed in immortalized p53-inactived sebocytes should not be transferred to the responses of primary human sebocytes, which were not used as physiological controls in the majority of experiments with immortalized sebocytes. However, the SGs of patients treated with isotretinoin in vivo exhibit a significant nuclear upregulation of p53 [179], a marked sebum reduction [360] and SG involution [361], most likely driven by p53-mediated apoptosis [215,216]. Immortalized sebocytes apparently mostly respond reasonably when studied under proliferative stimuli like exposure to IGF-1 and insulin but have an overlooked handicap with proapoptotic signaling, especially isotretinoin-regulated transcription. 

The final physiological fate of sebocytes in SGs is holocrine secretion [362], a unique DNase2-dependent mode of programmed cell death [363]. Notably, during *Drosophila* embryonic development, cell death eliminates 30% of primordial germ cells. Therefore, an intrinsic alternative cell death pathway is operative and mediated by the release of DNase2 from lysosomes, leading to nuclear translocation and subsequent DNA double-strand breaks [364,365]. Notably, the nuclear accumulation of DNase2 requires binding to the mitochondrial protein apoptosis-inducing factor (*AIFM1*), which is upregulated by p53 [366]. This sheds a new light on the mode of isotretinoin treatment which, via the upregulation of p53, may accelerate AIFM1/DNase2-dependent programmed cell death, a phenomenon which cannot be followed in p53-inactivated immortalized sebocytes. It is thus important for future research to focus on the transcriptomics of death pathways to attain deeper insights to improve our understanding of sebocyte biology as well as therapeutic responses [367]. Table 1 and Table 2 summarize p53- and FoxO1-controlled target genes which are critically involved in the pathogenesis of acne and isotretinoin’s mode of action.

## 9. Conclusions and Future Perspectives

The common inflammatory skin disease acne vulgaris, a highly prevalent disease in Western countries [368,369], is related to hereditary and multiple environmental factors [370,371], especially a Western-style diet (hyperglycemic carbohydrates; milk and dairy products) promoting insulin/IGF-1-driven nutrigenomic aberrations [372,373] (Figure 1). The observed changes in transcriptional regulation are primarily linked to downstream IGF-1/insulin/PI3K/AKT/mTORC1 signaling and androgen/AR/mTORC2/AKT signaling, which compromise the nuclear activities of p53, FoxO1 and FoxO3. The nuclear downregulation of FoxO1 results in the transactivation of AR, SREBF1, PPARγ and STAT3 and the attenuation of GATA6, critical transcription factors which promote sebocyte lipogenesis (increased sebum production), inflammation and infundibular hyperkeratinization (comedogenesis), respectively. The AKT-MDM2-mediated suppression of p53 modifies the expression of critical downstream target genes of p53.

In contrast, the most effective anti-acne agent, isotretinoin, attenuates all major pathogenic mechanisms of the disease. Since the introduction of isotretinoin for the treatment of cystic and conglobate acne in 1979 by Peck et al. [374], isotretinoin’s inhibitory mode of action on the pilosebaceous unit has remained a miracle. In the same year, p53 was detected [337].

After four decades, the current evidence, summarized in this review, allows for the conclusion that isotretinoin, via the upregulation of p53, FoxO1 and FoxO3, counteracts all major pathways in the pathogenesis of acne. Isotretinoin-induced sebocyte apoptosis with sebum suppression appears to be a result of the upregulation of the proapoptotic transcription factors p53 [179], FoxO1 and FoxO3 [176]. The chronic overexpression of p53 and FoxO1 may explain the depletion of BLIMP1(+) progenitor cells [164,168], a potential key mechanism explaining isotretinoin-induced SG involution.

However, forced p53/FoxO1/FoxO3 signaling also explains the adverse effects of isotretinoin treatment, including its teratogenicity (NCC apoptosis), ovarian reserve reduction (granulosa cell apoptosis), the risk of depression (reduced hypothalamic neurogenesis), dry skin (the upregulation of *AQP3*), hypertriglyceridemia (the upregulation of *APOB*) and the risk of intracranial hypertension (upregulation of *AQP1*).

p53-inactivation by SV40 large T antigen in immortalized sebocytes (SZ95; SEB-1) is an overlooked critical limitation of these in vitro sebocyte models for acne research and certain pharmacological studies that depend on adequate death signaling pathways. After a period of enthusiasm, it is apparently time to return to primary sebocytes and in vivo studies. After four decades since the introduction of isotretinoin for acne therapy, we are beginning to understand its mode of action at the level of transcriptomic regulation. In the future, the role of STAT3 in the pathogenesis of acne deserves a closer investigation, as STAT3 inhibits the expression of p53 [375] and interacts with FoxO1 [26]. Another highly interesting research question should be answered: Does the upregulation of p53 by isotretinoin accelerate the process of programmed sebocyte death via holocrine secretion?

Finally, it should be realized that the reported transcriptomic deviations exhibit a significant overlap with the transcriptomic landscape observed in cancers common in Western civilization, especially prostate and breast cancer [376,377,378]. It is thus not surprising that epidemiological studies reported an association between severe acne during adolescence and later risks of prostate cancer [379,380] and breast cancer [381], respectively. In contrast, the transcriptional changes induced by isotretinoin are useful in the prevention and therapy of various cancers [382,383,384,385,386].

## Figures and Tables

**Figure 1 cells-12-02600-f001:**
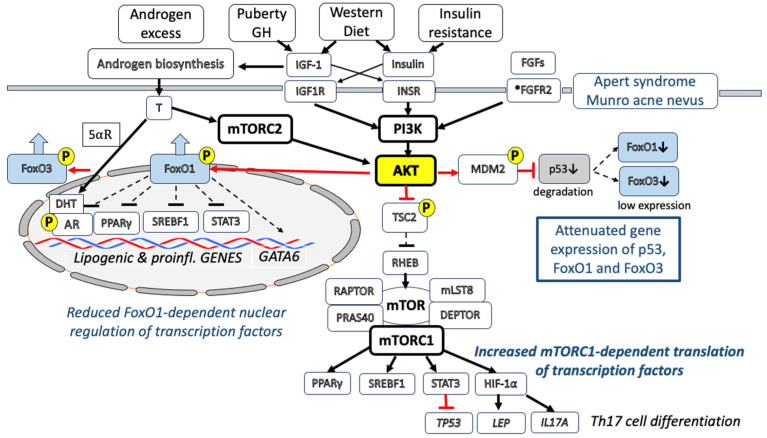
Disturbed transcriptional regulation in acne vulgaris. Growth factors and androgens overstimulate the kinase AKT (protein kinase B) in acne vulgaris. Increased puberty-mediated insulin-like growth factor 1 (IGF-1) signaling increases the activity of phosphoinositide 3-kinase (PI3K), which phosphorylates and activates AKT. Increased IGF-1/insulin signaling due to a Western diet (hyperglycemic carbohydrates; milk and dairy products), as well as fibroblast growth factor receptor 2 (FGFR2) gain-of-function mutations (Apert syndrome and Munro acne nevus), further augment the activation of AKT. IGF-1 stimulates adrenal and gonadal androgen biosynthesis and activates 5α-reductase (5α-R), converting testosterone (T) into dihydrotestosterone (DHT), the high-affinity ligand of androgen receptor (AR). Androgens activate mechanistic target of rapamycin complex 2 (mTORC2), which also phosphorylates and activates AKT. Thus, IGF-1 and androgens maintain a synergistic crosstalk, resulting in the activation of AKT. The AKT-mediated phosphorylation of AR mediates its nuclear transfer. In contrast, the AKT-mediated phosphorylation of the transcription factors forkhead box O1 (FoxO1) and forkhead box O3 (FoxO3) promotes their export from the nucleus into the cytoplasm, reducing their nuclear activity. FoxO1 is a suppressive nuclear coregulator of AR, peroxisome proliferator-activated receptor gamma γ (PPARγ), sterol regulatory element-binding transcription factor 1 (SREBF1) and signal transducer and activator of transcription 3 (STAT3) but promotes the expression of GATA-binding protein 6 (GATA6), the key regulatory transcription factor in infundibular keratinocytes. Thus, the loss of nuclear FoxO1 activity enhances the expression of lipogenic genes (activated by AR, PPARγ and SREBF1) and proinflammatory genes (activated by STAT3) and attenuates the expression of GATA6. The AKT-mediated phosphorylation of mouse-double minute 2 (MDM2) enhances the degradation of the transcription factor p53, resulting in the reduced p53-mediated expression of FoxO1, FoxO3 and other p53 target genes. The AKT-mediated phosphorylation of tuberin (TSC2) reduces its negative impact on Ras homolog protein enriched in brain (RHEB), the key activator of mechanistic target of rapamycin complex 1 (mTORC1). Activated mTORC1 stimulates the protein translation of the transcription factors PPARγ, SREBF1, STAT3 and hypoxia-inducible factor 1α (HIF-1α). STAT3 is a negative regulator of *TP53*, whereas HIF-1α stimulates the expression of leptin (*LEP*) and interleukin 17A (*IL17A*) and Th17 cell differentiation.

**Figure 2 cells-12-02600-f002:**
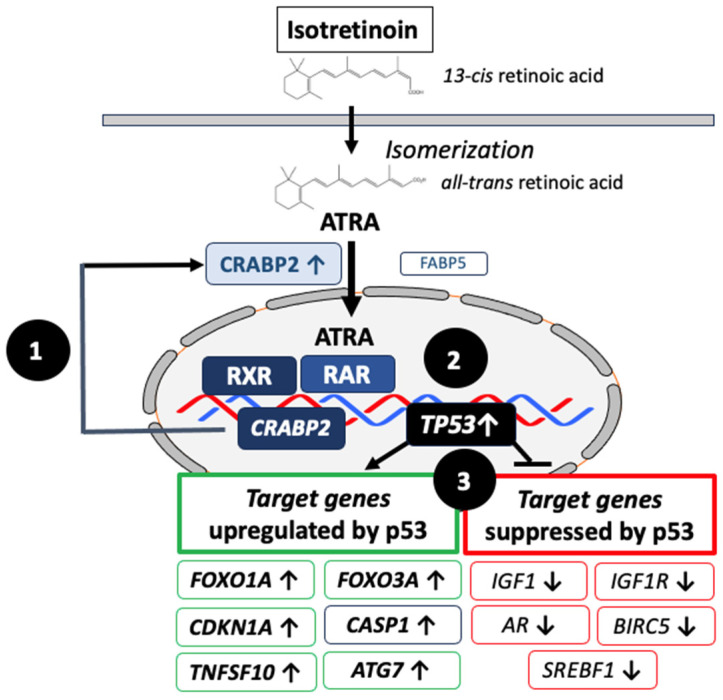
The isotretinoin-induced upregulation of p53. After the cellular uptake of isotretinoin, *13-cis* retinoic acid is isomerized to *all-trans* retinoic acid (ATRA). The further transport of ATRA depends on the cellular abundance of cellular retinoic-acid-binding protein 2 (CRABP2) which, in contrast to fatty-acid-binding protein 5 (FABP5), transfers ATRA to retinoic acid receptors (RARs), promoting the expression p53. The first response (1) to isotretinoin requires the ATRA-mediated upregulation of CRABP2 itself via the activation of RAR/RXR heterodimers. An enhanced CRABP2-mediated transfer of ATRA to a RAR may then enforce the second response to ATRA (2), which upregulates the expression of the transcription factor p53. Finally, in a third wave, p53 (3) upregulates target genes promoting cell cycle arrest (*CDKN1A*), autophagy (*ATG7*) and apoptosis (*FOXO1A*, *FOXO3A*, *CASP1* and *TNFSF10*) but suppresses genes involved in growth factor signaling (*IGF1* and *IGF1R*), androgen signaling (*AR*), cell survival (*BIRC5*) and lipid biosynthesis (*SREBF1*).

**Figure 3 cells-12-02600-f003:**
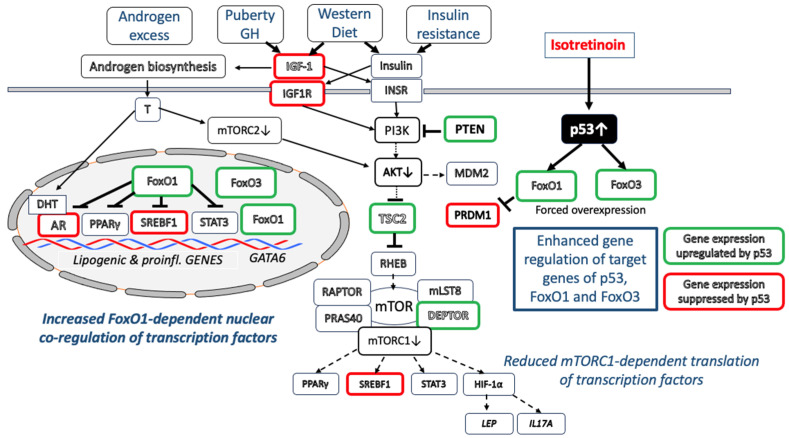
The isotretinoin-induced overexpression of p53 counteracts the dysregulated transcriptomics of acne vulgaris. Reduced levels of gene expression of insulin-like growth factor 1 (*IGF1*) and insulin-like growth factor 1 receptor (*IGF1R*) reduce the activation of phosphoinositide 3-kinase (PI3K), which is further suppressed by the upregulated expression of phosphatase and tensin homolog (*PTEN*). Reduced IGF-1/IGF1R signaling also attenuates androgen biosynthesis, resulting in reduced mTORC2-mediated activation of AKT. The increased expression of tuberin (*TSC2*) suppresses Ras homolog protein enriched in brain (RHEB), thereby reducing the activity of mTORC1. mTORC1 and mTORC2 are further inhibited via the induced expression of DEP-domain containing mTOR-interacting protein (*DEPTOR*), a natural inhibitor of both mTORC1 and mTORC2. Reduced mTORC1 results in the impaired protein translation of peroxisome proliferator activated receptor γ (PPARγ), sterol regulatory element-binding transcription factor 1 (SREBF1), signal transducer and activator of transcription 3 (STAT3) and hypoxia-inducible factor 1α (HIF-1α). The forced nuclear expression of FoxO1 inhibits the transcriptional activity of androgen receptor (AR), PPARγ, SREBF1 and STAT3 but induces the expression of *GATA6*. Thus, isotretinoin suppresses overactive growth factor and androgen signaling in acne patients but enhances GATA6 signaling, the key transcription factor controlling infundibular homeostasis. The forced overexpression of p53, FoxO1 and FoxO3 augments proapoptotic signaling, explaining isotretinoin’s desired pharmacological mode of action (sebum suppression via sebocyte apoptosis) as well as its major adverse effects, especially its teratogenicity (neural crest cell apoptosis). Notably, p53 and FoxO1 inhibit the expression of *PRBM1* (BLIMP1), a key marker of progenitor cells.

**Table 1 cells-12-02600-t001:** p53 target genes whose expression is either down- or upregulated by p53.

Gene	Gene Name	References
*AR*	Androgen receptor ↓	[80]
*IGF1*	Insulin-like growth factor 1 ↓	[191]
*IGF1R*	Insulin-like growth factor I receptor ↓	[193]
*BIRC5*	Baculoviral IAP repeat-containing protein 5 ↓	[220]
*SREBF1*	Sterol regulatory element-binding transcription factor 1 ↓	[350]
*CDKN1A*	Cyclin-dependent kinase inhibitor 1A (p21) ↑	[117]
*FOXO1A*	Forkhead box O1A ↑	[177]
*FOXO3A*	Forkhead box O3A ↑	[178]
*PTEN*	Phosphatase and tensin homog ↑	[195]
*PRKAB1*	Protein kinase, AMP-activated, noncatalytic, β-1 ↑	[195]
*TSC2*	TSC complex subunit 2 ↑	[195]
*DEPDC6*	DEP domain-containg protein 6 (DEPTOR) ↑	[197]
*TNFSF10*	Tumor necrosis factor ligand superfamily, member 10 ↑	[217]
*ATG7*	Autophagy-related 7 ↑	[231]
*APOB*	Apolipoprotein B ↑	[307]
*AQP3*	Aquaporin 3 ↑	[318]
*AQP1*	Aquaporin 1 ↑	[326]
*CASP1*	Caspase 1 ↑	[332]
*ALOX5*	Arachidonate 5-lipoxigenate ↑	[359]
*AIFM1*	Apoptosis-inducing factor, mitochondria associated ↑	[366]

**Table 2 cells-12-02600-t002:** Genes whose expression is either downregulated or upregulated by FoxO1.

Gene Symbol	Gene Name	References
*AR*	Androgen receptor ↓	[23]
*SREBF1*	Sterol regulatory element-binding transcription factor 1 ↓	[24]
*PPARG*	Peroxisome proliferator-activated receptor gamma γ ↓	[25]
*STAT3*	Signal transducer and activator of transcription ↓	[26]
*PRDM1*	PR domain-containing protein 1 (BLIMP1) ↓	[168]
*FSHB*	FSH β-polypeptide downregulation ↓	[289]
*LHB*	β-subunit of LH downregulation ↓	[290]
*GATA6*	GATA-binding protein 6 ↑	[112,114]
*FASLG*	Fas ligand ↑	[278]
*MTTP*	Microsomal triglyceride transfer protein ↑	[308]
*APOC3*	Apolipoprotein C-III ↑	[309,310]

## Data Availability

Not applicable.

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
