# Peer review of "Acne Transcriptomics: Fundamentals of Acne Pathogenesis and Isotretinoin Treatment"

_cells, 2023, doi:10.3390/cells12222600_

Round 1

Reviewer 1 Report

Comments and Suggestions for Authors

Acne Transcriptomics: Fundamentals in Acne Pathogenesis and Isotretinoin Treatment

Bodo C. Melnik

This is a narrative review, well-written and well-illustrated.

The possible contribution of AI is not mentioned.

There are two major flaw which strongly diminish the credibility of the whole.

 Flaw 1

Example LINE 272 273

Isotretinoin/ATRA induced overexpression of p53 and FoxO1 with associated attenuation 272 of PI3K-AKT-mTORC1 signaling may stabilize infundibular GATA6 in acne, a potential 273 key mechanism of comedolysis.

Only oral isotretinoin 13cisRA is active in acne, through atrophy of sebaceous gland, putatively due to sebocyte apoptosis induced by one of 13cisRA metabolites. (ref 170 is misleading in this issue)

Oral tretinoin all-trans RA does not induce these effects!

Major efforts by the industry (Roche & others) to find another retinoid (including ATRA) showing the same effect as 13cisRA were unsuccessful for the last 25 years.

It is therefore detrimental for the upcoming generations to dissimulate this evidence and try to sell 13-cisRA & ATRA as being “like in equilibrium” for all the putative effects deduced from a catalog of experiments or extrapolations.

The transcriptomic of RA /RAR RXR-dependent and relevant effects of retinoids in acne remain to be found. The author should outline that, any sound in vitro study with ATRA or 13cisRA should show, by appropriate controls the % of each resulting in the culture.

Flaw 2

The author does not mention lipid droplets proteins which have recently been shown to contribute to free fatty acids decrease in clinical human studies, contrary to most of other in vitro speculations emphasized in this review.

As well recent data on the role of sebaceous gland stem cells, which is now commonly considered as key in maintenance of homeostasis is even not mentioned whereas there are many overlap with the trancriptomics reviewed here.

This unfortunately gives to this review an auto-centered look.

Author Response

Response to R1

I thank reviewer 1 for all thoughtful comments, critical and most helpful remarks, which substantially improved the manuscript.

Remark 1

“Isotretinoin/ATRA signaling”

I have added a new subsection 7.1 entitled: “Isotretinoin´s in Vitro- Versus in Vivo Gene Regulatory Effects”. In this section, I compare the in vitro and in vivo findings supporting my preferred view of isotretinoin/ATRA signaling as the major mechanism for isotretinoin´s upregulation of p53, FoxO1 and FoxO3. For further details, please refer to subsection 7.1 in the revised manuscript.

Remark 2

“Lipid droplet proteins”

I have provided a further subsection 7.6 entitled: “Perilipin 2-Mediated Suppression of Comedogenesis”. I found convincing relations between p53 to PLN2 expression, very well explaining the findings of Sorg et al. For details, please refer to this subsection 7.6.

“Role of sebaceous gland stem cells”

I am very thankful for this point. In fact, translational evidence supports the view that the expression of BLIMP1 is controlled by p53 and FoxO1, providing a very exciting new aspect for the manuscript. This aspect is covered in the new subsection 7.2, entitled “Isotretinoin´s Potential Impact on Sebocyte Stem and Progenitor Cells”. For further details, please refer to this subsection.

Figures 2 and 3 have been improved accordingly.

Many thanks for all helpful remarks.

Reviewer 2 Report

Comments and Suggestions for Authors

The article is well-written and figures are helpful. The only critique is that this review article is a "re-hash" of the author's prior publications focused on the same hypothesis. It is not clear what additional, new insights are garnered in this review that have not already been discussed by the author (moderate comment).

Author Response

Response to R2

The reviewer criticizes that the review does contain already published information and no new aspects.

Therefore, I have added new data on p53/FoxO1-mediated suppression of BLIMP1, the key marker of progenitor cells. Furthermore, a new section addresses the role of perilipin 2 in comedogenesis and its relation to p53 signaling. In addition, I found more evidence for p53-regulated expression of AQP1 and p53/FoxO1 signaling in the regulation of hypothalamic neurogenesis.

I a new section, I have compared the in vitro versus in vivo findings of isotretinoin signaling in sebocytes and human sebaceous glands with special emphasis on the time course of the signaling events, explaining contradictory views in the literature.

Many thanks for these remarks and general appreciation of the manuscript.

Reviewer 3 Report

Comments and Suggestions for Authors

Acne vulgaris is a common inflammatory skin disease associated with hormonal changes in puberty, genetics, and environmental factors like diet. It involves increased sebum production, abnormal keratinization, Cutibacterium acnes bacteria, and inflammation.

The pathogenesis involves overactivation of the IGF-1/insulin and androgen signaling pathways, which converge on activating the kinase AKT. This leads to inhibition of the transcription factors p53, FoxO1 and FoxO3.

Inhibition of FoxO1 increases activity of lipogenic/inflammatory transcription factors like AR, SREBF1, PPARγ, STAT3, but reduces GATA6 which is important for normal keratinization.

p53 normally suppresses growth signaling pathways and promotes apoptosis. Its inhibition dysregulates cell proliferation and survival.

Isotretinoin is an effective treatment for severe acne. It upregulates p53, FoxO1, and FoxO3 expression. This counteracts the exaggerated growth factor and androgen signaling in acne.

Upregulation of these pro-apoptotic transcription factors explains isotretinoin's sebum suppressive effect via increased sebocyte apoptosis. But it also contributes to adverse effects like teratogenicity and depression risk.

Immortalized sebocyte cell line models used in research have limitations due to inactivation of p53 signaling. Studies should focus more on in vivo models and primary sebocytes.

Understanding isotretinoin's effects on transcriptional regulation of apoptotic pathways provides insights into its therapeutic mechanism and side effects. More research is needed on crosstalk of key transcription factors like p53 and STAT3.

I like this review very much, about Acne. Here's my suggestion. Add some basics research in acne.

1.My suggestion is to add two more papers to elaborate the basic research. Recently, there is a paper on hair follicles updating sebaceous glands(Han, iscience 2023)(Natalia A. Veniaminova,cell reports,2023). I thought adding to your essay would flesh out your topic. Acne is related to sebaceous gland. And this is most recent paper about sebaceous gland mechanisms.

Author Response

Response to R3

I thank reviewer 3 for the great appreciation of the manuscript (5 stars points for all marks).

As requested, I have updated the literature and provide three new sections offering exciting new insights:

7.1 entitled: “Isotretinoin´s in Vitro- Versus in Vivo Gene Regulatory Effects”.

7.2, entitled “Isotretinoin´s Potential Impact on Sebocyte Stem and Progenitor Cells”.

7.6 entitled: “Perilipin 2-Mediated Suppression of Comedogenesis”.

Many thanks for the great appreciation of the work.

Round 2

Reviewer 1 Report

Comments and Suggestions for Authors

The author has interestingly answered to the questions by including news informations and integrating these in his own way of anaylsing a complex area. This may be a good basis for further discussions and very informative for readers